# Learning Inter-Atomic Potentials without Explicit Equivariance

## Abstract

Accurate and scalable machine-learned inter-atomic potentials (MLIPs) are essential for molecular simulations ranging from drug discovery to new material design. Current state-of-the-art models enforce roto-translational symmetries through equivariant neural network architectures, a hard-wired inductive bias that can often lead to reduced flexibility, computational efficiency, and scalability. In this work, we introduce **TransIP**: **Trans**former-based **I**nter-Atomic **P**otentials, a novel training paradigm for interatomic potentials achieving symmetry compliance without explicit architectural constraints. Our approach guides a generic non-equivariant Transformer-based model to learn $SO(3)$-equivariance by optimizing its representations in the embedding space. Trained on the recent Open Molecules (OMol25) collection, a large and diverse molecular dataset built specifically for MLIPs and covering different types of molecules (including small organics, biomolecular fragments, and electrolyte-like species), TransIP attains comparable performance in machine-learning force fields versus state-of-the-art equivariant baselines. Further, compared to a data augmentation baseline, TransIP achieves 40% to 60% improvement in performance across varying OMol25 dataset sizes. More broadly, our work shows that learned equivariance can be a powerful and efficient alternative to equivariant or augmentation-based MLIP models.

## 1 Introduction

Atomistic simulations are a fundamental task in chemistry and materials science (Zhang et al., 2018; Deringer et al., 2019), with Density Functional Theory (DFT) serving as a basis for accurately calculating interatomic forces and energies. However, the utility of DFT is severely restricted by its computational costs, which typically scale cubically with system size, rendering large-scale or long-timescale simulations intractable. This has motivated machine-learned interatomic potentials (MLIPs) to overcome this limitation by learning the potential energy surface from data, offering orders-of-magnitude speed-ups compared to DFT calculations (Noé et al., 2020; Batzner et al., 2022; Batatia et al., 2022; Jacobs et al., 2025; Leimeroth et al., 2025).

Equivariant neural networks have become a central paradigm for MLIPs due to their ability to encode the three-dimensional structure of molecular graphs (Anderson et al., 2019; Thölke & Fabritiis, 2022; Liao et al., 2024a; Fu et al., 2025). These architectures are designed to explicitly respect roto-translational symmetries (SE(3) equivariance) by construction, often employing compute-intensive mechanisms like spherical harmonics or equivariant message passing (Fuchs et al., 2020; Passaro & Zitnick, 2023a; Liao & Smidt, 2023; Maruf et al., 2025). However, due to the design difficulties and limited expressive power of these architectures (Joshi et al., 2023; Cen et al., 2024), a recent trend in predictive and generative modeling is to use unconstrained models when enough data is available (Wang et al., 2024; Abramson et al., 2024; Zhang et al., 2025a; Joshi et al., 2025).

In this paper, we introduce **TransIP** (**Trans**former-based **I**nteratomic **P**otentials), a training paradigm that achieves molecular symmetry for interatomic potentials *without* imposing architectural $SO(3)$ constraints. TransIP steers a standard transformer toward $SO(3)$ equivariance via an additional contrastive objective, allowing the model to retain the scalability and hardware efficiency of attention mechanisms while learning symmetry from data.

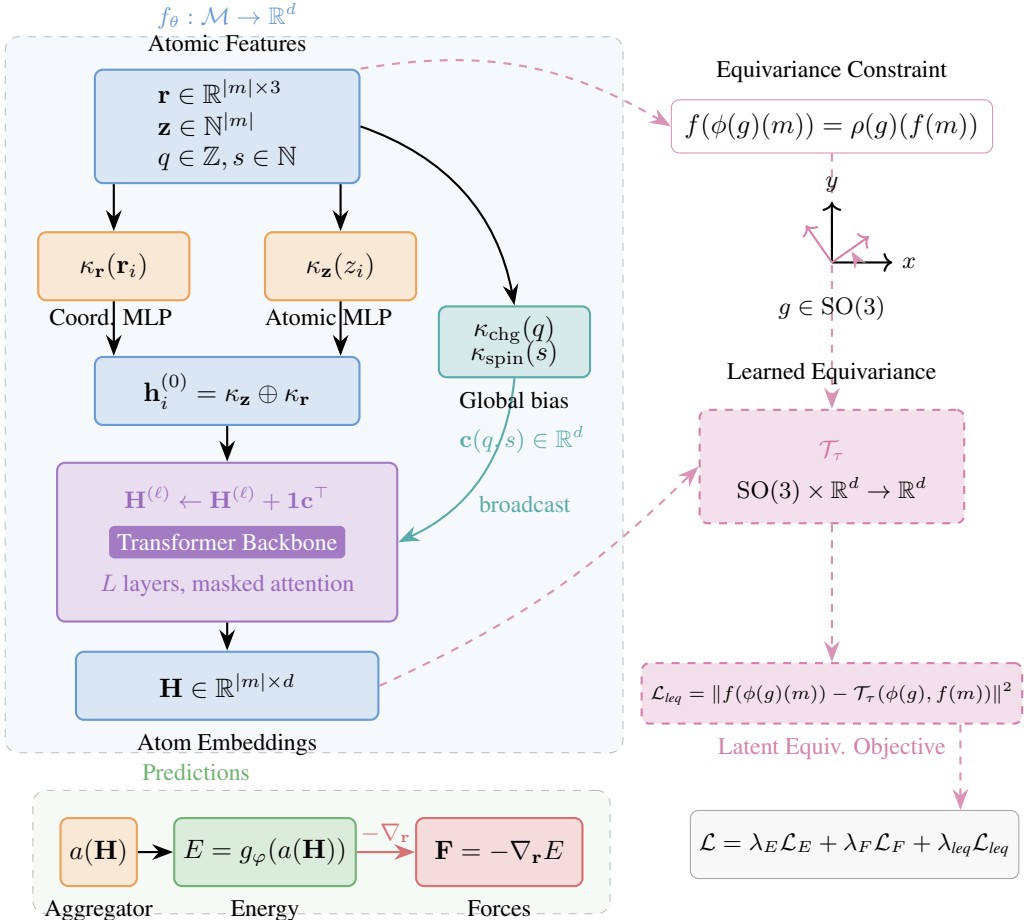

Figure 1: TransIP: Transformer-based Interatomic Potentials.

Our contributions are as follows:

- We propose an MLIP training pipeline with a general transformer-based model to obtain $SO(3)$ equivariance through training rather than hard-wired equivariant layers.

- We introduce an architecture-agnostic contrastive loss function that promotes $SO(3)$ equivariance in the embedding space of an unconstrained model. By aligning latent features across $SO(3)$ transformations in the model's backbone, we show that TransIP scales better across different datasets and model sizes compared to traditional data augmentation techniques.

- On a diverse molecular benchmark, Open Molecules 25 (Levine et al., 2025) (that includes small organics, biomolecular fragments, electrolyte-like species), we show that TransIP outperforms data augmentation techniques and achieves comparable performance versus current state-of-the-art MLIP baselines.

## 2 SYMMETRY IN EMBEDDING SPACE

### 2.1 PROBLEM FORMULATION

**Molecular representations.** Let $\mathcal{M}$ denote the space of molecular configurations. Each molecule $m \in \mathcal{M}$ is represented by atomic features $\mathbf{x} = (\mathbf{r}, \mathbf{z}, q, s)$, where $\mathbf{r} \in \mathbb{R}^{|m| \times 3}$ are atomic coordinates, $\mathbf{z} \in \mathbb{N}^{|m|}$ are atomic numbers, $q \in \mathbb{Z}$ is the total molecular charge, and $s \in \mathbb{N}$ is the spin multiplicity, with $|m|$ denoting the number of atoms in molecule $m$.

Our goal is to learn an embedding function $f_\theta : \mathcal{M} \to \mathbb{R}^d$ that maps molecular configurations to a $d$-dimensional latent space, and a prediction function $g_\varphi : \mathbb{R}^d \to \mathbb{R}$ that acts in the embedding space $\mathbb{R}^d$ and outputs molecular properties (e.g., energy). Both $f_\theta$ and $g_\varphi$ are neural networks parameterized by $\theta$ and $\varphi$, respectively.

**Symmetry groups.** We define a symmetry group $G$ that acts on a set $\mathcal{X}$ as a group of bijective functions from $\mathcal{X}$ to itself, and the group operation is function composition. We say a function $f$ is *equivariant* w.r.t. the group $G$ if for every transformation $g \in G$ and every input $x \in X$,

$$f(\phi(g)(x)) = \rho(g)(f(x)) \tag{1}$$

The group representations $\phi$ and $\rho$ specify how we apply the elements of the group $G$ on input and output data. As a concrete case, we can define $G$ as a rotation group $\mathrm{SO}(3)$ over molecular configurations $\mathcal{M}$, with $g \in \mathrm{SO}(3)$ representing an element of $G$ that acts on a molecule $m$ by rotating the coordinates of each atom in 3D space. Formally, for a molecule $m = (\mathbf{r}, \mathbf{z}, q, s)$ with coordinates $\mathbf{r} = (\mathbf{r}_1, \ldots, \mathbf{r}_{|m|})$, $\mathbf{r}_i \in \mathbb{R}^3$, the input action rotates each atom:

$$\big(\phi(g)\, m\big) = \big((R\mathbf{r}_1, \ldots, R\mathbf{r}_{|m|}),\ \mathbf{z},\ q,\ s\big).$$

Here $R$ is a $3 \times 3$ rotation matrix (orthogonal with $\det R = 1$); $\mathbf{z}, q, s$ are unchanged. An associated output representation rotates vector-valued quantities—e.g., for forces $\mathbf{F} = (\mathbf{F}_1, \ldots, \mathbf{F}_{|m|})$, $\rho(g)\mathbf{F} = (R\mathbf{F}_1, \ldots, R\mathbf{F}_{|m|})$—while scalar outputs such as energies remain invariant, $\rho(g)E = E$.

## 2.2 Implicit Equivariance in Embedding Space

We seek an embedding function $f$ that behaves equivariantly with respect to the symmetry group $G$, meaning there exists a transformation $\rho(g) : \mathbb{R}^d \to \mathbb{R}^d$ such that:

$$f(\phi(g)(m)) = \rho(g)(f(m)) \quad \forall g \in G, m \in \mathcal{M} \tag{2}$$

Common approaches enforce equivariance constraints through specialized architectures. Instead, we want the embedding function $f$ to learn symmetry without equivariance constraints. However, with $G$ being the rotation group $\mathrm{SO}(3)$ on $\mathcal{M}$ and the output of $f$ being a high-dimensional vector, there is no direct representation of $\rho(g)$ to act in the space of $\mathbb{R}^d$. Thus, rather than specifying $\rho(g)$ analytically, we propose to learn the group transformation on an embedding vector in $\mathbb{R}^d$ using a neural network $\mathcal{T}_\tau : \mathrm{SO}(3) \times \mathbb{R}^d \to \mathbb{R}^d$ parameterized by $\tau$. $\mathcal{T}$ can be understood as a non-linear function that learns the group action implicitly on a latent vector, by providing the group representation on the input data.

# 3 Learning Inter-Atomic Potentials without Explicit Equivariance

In this section, we introduce our training framework: TransIP (Transformer-based Inter-atomic Potentials), a new approach that achieves $\mathrm{SO}(3)$-equivariance through learned transformations in an embedding space without explicit equivariance constraints. Our method, illustrated in Figure 1, consists of three key components: (i) an unconstrained Transformer backbone that processes molecular configurations, (ii) a learned transformation network that performs group actions in the embedding space, and (iii) a contrastive objective that enforces latent equivariance (equiv.) during training.

## 3.1 TransIP: Transformer-based Interatomic Potentials

**Atom as tokens.** We model each molecule as a variable-length sequence of tokens, where each token represents an atom. Unlike conventional graph neural networks that construct edges based on distance cutoffs or neighbours' atoms, we process all atoms within a molecule through self-attention, bounded by a maximum context length $N_{\mathrm{ctx}}$. For batch processing, we use padding masks to prevent cross-molecule attention, ensuring each molecule is processed independently.

In addition, we apply rotary position embeddings (RoPE) (Su et al., 2023) to the queries $\mathbf{q}_i \in \mathbb{R}^{d/h}$ and keys $\mathbf{k}_j \in \mathbb{R}^{d/h}$ of each attention head, where $i, j$ denote the sequence positions of atoms within

a molecule, $d$ is the model dimension, and $h$ is the number of attention heads. The attention weights are computed as:

$$\tilde{\mathbf{q}}_i = \text{RoPE}(\mathbf{q}_i, i), \quad \tilde{\mathbf{k}}_j = \text{RoPE}(\mathbf{k}_j, j)$$

$$\alpha_{ij} = \text{softmax}\left(\frac{\tilde{\mathbf{q}}_i^\top \tilde{\mathbf{k}}_j}{\sqrt{d/h}} + m_{ij}\right)$$

where $\text{RoPE}(\cdot, \cdot)$ is the rotary position encoding operator, and $m_{ij} \in \{0, -\infty\}$ is the attention mask that blocks padding tokens and enforces within-molecule attention. This approach eliminates the need for explicit distance cutoffs while maintaining flexibility in modeling molecular interactions.

**Transformer Backbone.** We implement the embedding function $f_\theta : \mathcal{M} \to \mathbb{R}^d$ as a Transformer encoder that processes atom-level tokens. Each atom $i$ is initialized with a token representation:

$$\mathbf{h}_i^{(0)} = \kappa_{\mathbf{z}}(z_i) \oplus \kappa_{\mathbf{r}}(\mathbf{r}_i)$$

where $\kappa_{\mathbf{z}} : \mathbb{N} \to \mathbb{R}^d$ and $\kappa_{\mathbf{r}} : \mathbb{R}^3 \to \mathbb{R}^d$ are learnable MLPs that embed atomic numbers and centered coordinates (with $\mathbf{r}_i \leftarrow \mathbf{r}_i - \frac{1}{|m|}\sum_j \mathbf{r}_j$), and $\oplus$ denotes concatenation. These tokens are processed through $L$ Transformer layers with masked self-attention within each molecule, producing final per-atom embeddings $\mathbf{H} = [\mathbf{h}_1, \ldots, \mathbf{h}_{|m|}]^\top \in \mathbb{R}^{|m| \times d}$.

**Global Molecular Properties.** Following Levine et al. (2025), we incorporate global molecular properties (total charge $q$ and spin multiplicity $s$ of a molecule $m$) through learnable embeddings, and form a graph-level bias:

$$\mathbf{c}(q, s) = \kappa_{\text{chg}}(q) + \kappa_{\text{spin}}(s) \in \mathbb{R}^d$$

where $\kappa_{\text{chg}}$ and $\kappa_{\text{spin}}$ are learnable embedding functions for charge and spin, respectively. This global bias is broadcast-added at each Transformer layer: $\mathbf{H}^{(\ell)} \leftarrow \mathbf{H}^{(\ell)} + \mathbf{1}\mathbf{c}(q, s)^\top$.

**Energy and Force Predictions.** For molecular property prediction, we employ a permutation-invariant aggregator $a : \mathbb{R}^{|m| \times d} \to \mathbb{R}^d$ followed by an energy prediction head $g_\varphi : \mathbb{R}^d \to \mathbb{R}$:

$$E_\varphi(m) = g_\varphi(a(\mathbf{H}))$$

Forces are computed as conservative gradients of the energy with respect to atomic positions:

$$\mathbf{F}(m) = -\nabla_{\mathbf{r}} E_\varphi(m) \in \mathbb{R}^{|m| \times 3}$$

## 3.2 LEARNED LATENT EQUIVARIANCE

**Transformation Network.** We propose a transformation network $\mathcal{T}_\tau : \text{SO}(3) \times \mathbb{R}^d \to \mathbb{R}^d$ that learns how group actions (e.g., rotations) act on molecular embeddings. We implement $\mathcal{T}_\tau$ as a multilayer perceptron that takes as input the group representation in the input domain $\phi(g)$ and the molecular embedding $f(m)$. Formally,

$$\mathcal{T}_\tau(\phi(g), f(m)) = \text{MLP}_\tau([\phi(g), f(m)])$$

where $[\cdot, \cdot]$ denotes concatenation and $\text{MLP}_\tau$ is a multilayer perceptron with parameters $\tau$.

**Contrastive Objective for Latent Equivariance:** To learn the molecular symmetry without architectural constraints, we define our latent equivariance loss as:

$$\mathcal{L}_{leq}(\phi(g), m, f, \mathcal{T}) = \|f(\phi(g)(m)) - \mathcal{T}_\tau(\phi(g), f(m))\|^2 \tag{3}$$

This loss encourages the embedding function $f$ to behave equivariantly with respect to the symmetry group $G$, as mediated by the transformation network $\mathcal{T}_\tau$. During training, we sample a molecule $m$ from the dataset and a rotation element $g$ uniformly from $\text{SO}(3)$ and minimize the expected latent loss:

$$\min \mathbb{E}_{m \sim \mathcal{M}, g \sim \text{SO}(3)}[\mathcal{L}_{leq}(\phi(g), m, f, \mathcal{T})] \tag{4}$$

### 3.3 TRAINING OBJECTIVE

Our training objective combines three complementary losses for accurate prediction of energy and forces as well as implicitly learning molecular symmetry.

**Prediction Losses.** For energy and force predictions, we use:

$$\mathcal{L}_E = \frac{1}{|m|}|E_\varphi(m) - E^\star| \quad \text{(per-atom mean absolute error (MAE))} \tag{5}$$

$$\mathcal{L}_F = \frac{1}{3|m|}\|\mathbf{F}(m) - \mathbf{F}^\star\|_F^2 \quad \text{(per-molecule mean squared error (MSE))} \tag{6}$$

where $E^\star$ and $\mathbf{F}^\star$ are ground-truth energies and forces, and $\|\cdot\|_F$ denotes the Frobenius norm. For energies, we use referenced targets as described by Levine et al. (2025).

**Combined Objective.** Training combines three weighted terms: (i) the latent equivariance target $\mathcal{L}_{leq}$ defined in Eq. 3; (ii) energy loss $\mathcal{L}_E$; and (iii) force loss $\mathcal{L}_F$. The total objective is

$$\mathcal{L}_{\text{total}} = \lambda_E \mathcal{L}_E + \lambda_F \mathcal{L}_F + \lambda_{leq} \mathcal{L}_{leq} \tag{7}$$

where $\lambda_E$, $\lambda_F$, and $\lambda_{leq}$ are hyperparameters for each loss. The optimal hyperparameters are given in Table 5 of Appendix A.

## 4 RELATED WORK

**ML Interatomic Potentials.** Using machine learning (ML) methods to predict energies and forces of different molecular systems and materials has been an active area of research (Schütt et al., 2017; Chmiela et al., 2022; Musaelian et al., 2023; Liao et al., 2024b; Yang et al., 2025). Due to the intricate 3D structures of atomistic systems, equivariant designs such as steerable convolution (Cohen & Welling, 2017; Brandstetter et al., 2022) and higher-order tensors (Thomas et al., 2018), as well as covariant representation (Anderson et al., 2019), have been essential backbones for modeling molecular systems. For example, Gasteiger et al. (2020); Klicpera et al. (2021) introduced equivariant directional message passing between pairs of atoms with a spherical harmonics representation. In contrast, Batzner et al. (2022) developed equivariant convolution with tensor-products and Batatia et al. (2022) built higher-order messages with equivariant graph neural networks (Satorras et al., 2021). Additionally, Passaro & Zitnick (2023b) reduced the computational complexity of SO(3) convolution and replaced it with SO(2) convolutions, which have been used as a backbone for MLIPs (Fu et al., 2025). More recently, Rhodes et al. (2025) presented Orb-v3 models with improved computational efficiency, built on Graph Network Simulators (Sanchez-Gonzalez et al., 2020).

**Unconstrained ML models.** While current-state-of-the-art MLIP models primarily rely on equivariant GNNs, unconstrained models are actively used in other domains. For example, integrating data augmentation via image transformations has been used in different vision tasks, from classification (Inoue, 2018; Dosovitskiy et al., 2021; Rahat et al., 2024) to segmentation (Negassi et al., 2022; Yu et al., 2023). For geometric data, the use of unconstrained models and diffusion Transformers (without explicit equivariance constraints) has been a recent trend in generative tasks, e.g., AlphaFold 3 for biomolecular structure prediction (Abramson et al., 2024) as well as molecular conformation and materials generation (Wang et al., 2024; Zhang et al., 2025a; Joshi et al., 2025). In contrast, several works have been introduced to overcome the limitations of strictly equivariant GNNs by enforcing symmetry via frame averaging over geometric inputs (Puny et al., 2022; Duval et al., 2023; Lin et al., 2024; Huang et al., 2024; Dym et al., 2024); learning canonicalization functions that map inputs to a canonical orientation before prediction (Kaba et al., 2022; Baker et al., 2024; Ma et al., 2024; Lippmann et al., 2025); or learning equivariance through data augmentation with molecule-specific graph-based architectures (Qu & Krishnapriyan, 2024; Mazitov et al., 2025). However, in this work, we demonstrate that an unconstrained general-purpose Transformer model can serve as a backbone for MLIPs, which replaces graph-based inductive biases with a scalable latent equivariance objective that implicitly learns equivariant features without explicit equivariance constraints.

## 5 EXPERIMENTAL SETUP

**Dataset.** We train and evaluate our proposed method **TransIP** on the Open Molecules 2025 (OMol25) collection (Levine et al., 2025), a large-scale molecular DFT dataset for ML interatomic potentials. OMol25 covers 83 atomic elements and diverse chemistries including: metal complexes, electrolytes, biomolecules, SPICE, neutral organic, and reactivity. It contains molecules from several datasets such as ANI-2X (Devereux et al., 2020), Transition-1X (Schreiner et al., 2022), ANI-1xBB (Zhang et al., 2025b), Orbnet Denali (Christensen et al., 2021), SPICE2 (Eastman et al., 2022; 2024), and Solvated Protein Fragments (Unke & Meuwly, 2019). Following Levine et al. (2025), we use the official 4M training split (3,986,754) and the out-of-distribution composition validation split *Val-Comp* (2,762,021). *Val-Comp* consists of molecules gathered from various datasets and domains, such as biomolecules, neutral organics, and metal complexes.

**Model Configurations.** We evaluate TransIP across three model scales: Small (14M parameters), Medium (85M parameters), and Large (302M parameters). All models use MLP-based coordinate embeddings and RoPE positional encodings. The transformation network $\mathcal{T}_\tau$ is a 2-layer MLP with GELU activations and $2d$ hidden dimension.

**Training Setup.** Using the standardized FAIRCHEM Python package (Shuaibi et al., 2025), we train TransIP on the OMol25 dataset using an AdamW optimizer with learning rate $5 \times 10^{-4}$, weight decay $10^{-3}$, and gradient norm clipping at 200. We use a cosine learning rate schedule with linear warmup over the first 1% of training, followed by cosine decay down to 1% of the initial lr. The loss weights are set to $\lambda_E = 5$ for energies and $\lambda_F = 15$ for forces. For the latent equivariance objective $\lambda_{leq}$, we sweep the values in $\{1, 5, 10, 100\}$ and selected $\lambda_{leq} = 5$ based on validation performance.

**Scalability Experiments.** We conduct three sets of experiments to assess TransIP's scaling behavior:

- **Data scaling**: We train the Small (14M parameter) model on three dataset sizes (1M, 2M, 4M molecules) for 5 epochs using 8 NVIDIA 80GB GPUs, comparing TransIP with learned equivariance against an unconstrained Transformer version with $SO(3)$ data augmentation (TransAug).

- **Model size scaling.** We compare TransIP and TransAug with different model sizes (Small/Medium/Large) trained on the same number of samples from the OMol25 4M dataset and report the evaluation metrics as a function of the processed number of atoms per second.

- **Extended training**: We train TransIP (Small) on the OMol25 4M dataset for 40 epochs using 64 NVIDIA 80GB GPUs to evaluate its performance against standardized equivariant baselines.

**Baselines.** We compare TransIP against: (i) an *unconstrained* TransIP variant trained with $SO(3)$ rotation augmentation to assess the impact of learned latent equivariance versus data augmentations, and (ii) state-of-the-art equivariant models on OMol25: eSCN (Fu et al., 2025) in small/medium configurations with both direct and energy-conserving force variants as well as GemNet-OC (Gasteiger et al., 2022).

**Evaluation metrics.** Following the OMol25 official benchmark, we report: Force MAE (meV/Å), Force cosine similarity, Energy per atom MAE (meV/atom), and Total energy MAE (meV). Detailed metric definitions are provided in Appendix A.4.

## 6 RESULTS AND DISCUSSION

### 6.1 SCALING DATA SIZE

To assess how performance scales with different training dataset sizes, we compare our latent equivariance-based model (TransIP) against an unconstrained baseline that uses $SO(3)$ data augmentation (TransAug). Both models use a (small) 14M parameter Transformer architecture. Given our tight compute budget, we train on 1M, 2M, and 4M OMol25 molecules for 5 epochs and report validation (*Val-Comp*) results.

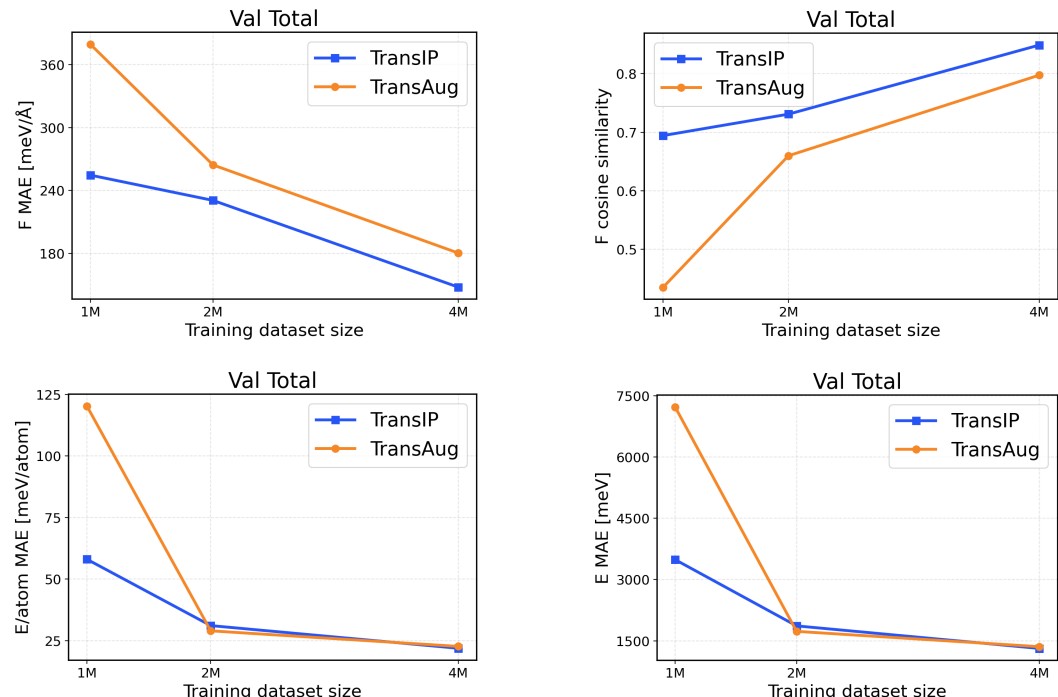

Figure 2: Val-Comp performance across different dataset sizes (1M / 2M / 4M): The top row presents force metrics, while the bottom row reports energy metrics.

**Performance in a limited data regime.** Figure 2 shows that TransIP delivers large gains when trained on 1M samples and outperforms TransAug across all evaluation metrics with a large margin on the total validation split. We also include the performance comparison for each molecule category in Appendix B. In Figure 2, the learned latent equivariance objective provides substantial improvements in force MAE (255 meV/Å vs 600 meV/Å MAE) and directional consistency (0.7 vs 0.44 force cosine similarity). Energy predictions also benefit from the latent equivariance objective, with TransIP achieving 58 meV/atom compared to TransAug's 120 meV/atom. These results suggest that learning equivariance in a latent space is a more effective scheme to incorporate molecular symmetry than data augmentation, particularly when training data is limited.

**Performance in a larger data regime.** As we scale to 2M and 4M molecules, both models (TransIP and TransAug) improve across the evaluation metrics. However, on larger datasets, TransIP still achieves better force MAEs and cosine similarity metrics compared to TransAug. This might indicate that the learned transformation network can successfully capture the geometric relationships necessary for accurate force predictions. Notably, energy prediction performance converges between the two at larger data scales, with both methods achieving comparable per-atom MAE values. This convergence suggests that while learned equivariance provides crucial benefits for force-related metrics in all data regimes, its advantages for energy prediction become less pronounced as the model can learn invariant energy representations from sufficient augmented data.

## 6.2 LEARNED LATENT EQUIVARIANCE

We investigate how learned equivariance affects the embedding space in relation to validation performance as the data scale increases. Figure 3 plots each metric against latent equiv. error for TransIP (Small) trained for 5 epochs on 1M, 2M, and 4M molecules (see Table 3 for a detailed definition of each model configuration).

**Lower latent equivariance error leads to better accuracy.** We found that the learned equiv. error serves as a strong predictor of model performance. Across all metrics, we observe a clear monotonic trend: lower equiv. error is associated with better performance (Figure 3). However, energy and force predictions respond differently to improvements in equivariance. Energy predictions

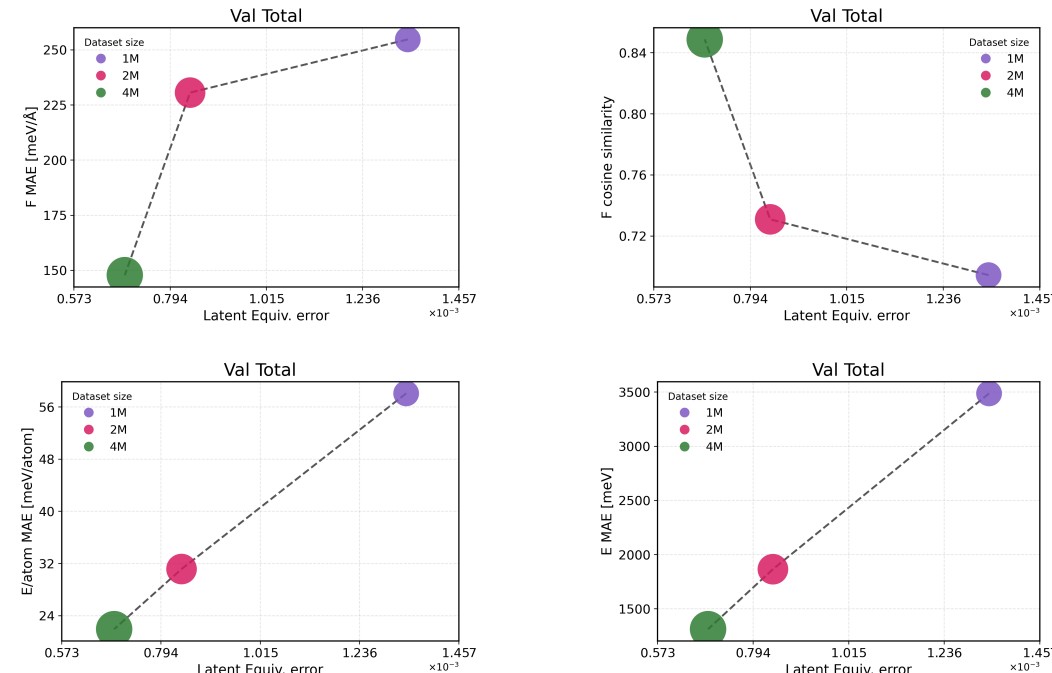

Figure 3: Latent equivariance (embedding) error versus validation performance. The top row reports force metrics, while the bottom row presents energy metrics.

show near-linear scaling with equiv. error, indicating that energy accuracy is directly limited by equivariance quality. This strong coupling aligns with energies being scalar invariants that depend primarily on learning correct symmetry-preserving features. In contrast, force predictions exhibit a two-regime behavior: initial improvements in equivariance (1M→2M) yield modest force improvements, while further tightening of equivariance (2M→4M) produces disproportionate gains. This might indicate that forces require both accurate equivariant features and sufficient data diversity to learn the energy landscape's geometry.

These results demonstrate that implicitly learning equivariance through our learned transformation network provides an efficient inductive bias, accelerating learning. The 48% reduction in equiv. error from 1M to 4M training examples translates to 40-60% performance improvements, being more efficient than what would be expected from data scaling alone.

**Learning equivariance leads to faster inference.** To measure the inference efficiency of our method, we compare TransIP and TransAug with different model sizes (Small/Medium/Large) trained on 4M samples and report the evaluation metrics as a function of the processed number of atoms per second. However, due to limited compute, we compare models under a *fixed training budget* (i.e., with the same number of samples), which is 10k, 25k, and 100k steps for our Small, Medium, and Large models, respectively.

From the results in Figure 4, we see that TransIP scales smoothly with parameter count despite limited training: As model size grows, performance improves across all metrics. In contrast, TransAug exhibits poorer scaling—larger models perform worse than smaller ones, with the Large model configuration yielding the lowest performance. This might indicate that augmentation alone does not provide a sufficiently informative and stable inductive bias for large-capacity models trained for molecular force field prediction.

### 6.3 ARCHITECTURAL EQUIVARIANCE VERSUS LEARNED EQUIVARIANCE

Table 1 compares the energy and force prediction performance of TransIP-S (Small) against TransAug-S (Small) as well as several well-known equivariant baselines for the OMol 2M Val-Comp evaluation dataset. The results of this comparison demonstrate that TransIP-S outperforms

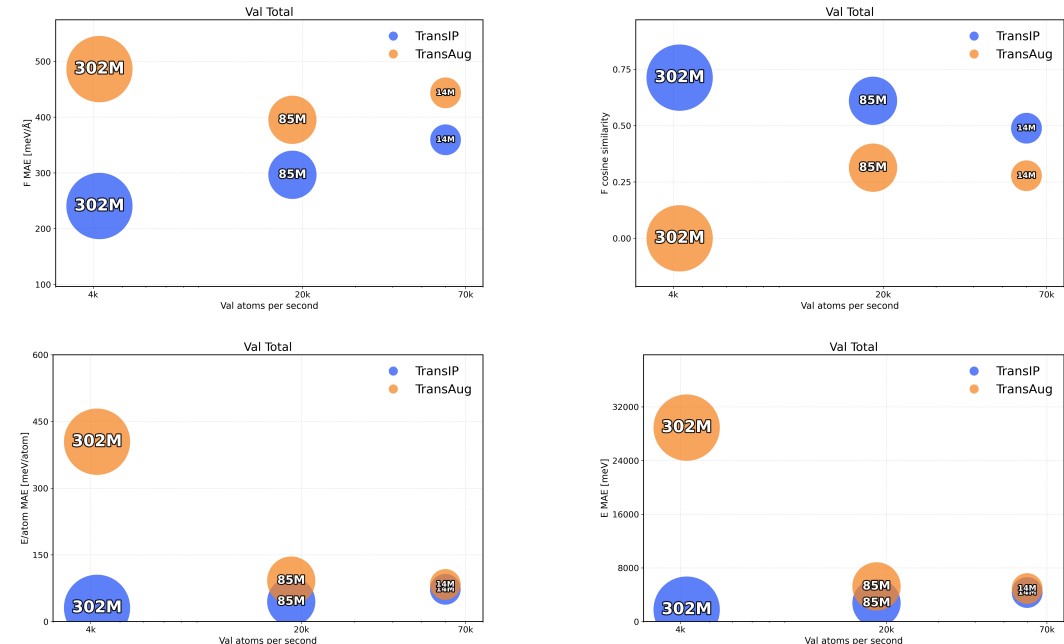

Figure 4: Validation total inference trade-off (atoms/s versus performance). The top row presents force metrics, while the bottom row represents energy metrics.

| | | Biomolecules | | Electrolytes | | Metal Complexes | | Neutral Organics | | Total | |
| --- | --- | --- | --- | --- | --- | --- | --- | --- | --- | --- | --- |
| **Model** | **Epochs** | **Energy ↓** | **Forces ↓** | **Energy ↓** | **Forces ↓** | **Energy ↓** | **Forces ↓** | **Energy ↓** | **Forces ↓** | **Energy ↓** | **Forces ↓** |
| eSEN-sm-d. | 80 | 0.88 | 8.12 | 1.93 | 12.64 | 3.37 | 40.44 | 2.16 | 20.17 | 2.19 | 13.01 |
| eSEN-sm-cons. | 80 | 0.86 | 6.17 | 1.61 | 11.16 | 2.72 | 35.33 | 1.50 | 16.92 | 1.89 | 11.10 |
| eSEN-md-d. | 80 | 0.47 | 3.38 | 1.18 | 6.51 | 2.53 | 27.31 | 1.21 | 9.26 | 1.32 | 6.78 |
| GemNet-OC-r6 | 80 | 0.40 | 5.84 | 1.39 | 9.37 | 2.74 | 33.60 | 1.88 | 16.55 | 1.41 | 9.83 |
| GemNet-OC | 80 | 0.25 | 5.20 | 1.04 | 8.42 | 2.66 | 32.76 | 1.64 | 15.59 | 1.13 | 8.98 |
| TransAug-S | 5 | 16.6 | 219.3 | 17.5 | 161.9 | 20.7 | 150.6 | 28.9 | 218.8 | 23.5 | 180.3 |
| TransIP-S | 5 | 17.3 | 181.1 | 15.9 | 129.6 | 18.5 | 132.5 | 23.5 | 165.0 | 22.3 | 146.6 |
| TransIP-S | 40 | 13.8 | 121.5 | 12.7 | 94.0 | 15.2 | 105.6 | 18.5 | 125.4 | 17.9 | 103.8 |
| TransIP-S | 80 | 10.5 | 103.2 | 10.2 | 82.4 | 13.8 | 96.1 | 16.0 | 111.4 | 14.2 | 90.1 |
| TransIP-M | 60 | 6.3 | 35.2 | 5.7 | 33.0 | 7.4 | 58.5 | 7.9 | 51.0 | 8.1 | 35.4 |

Table 1: Comprehensive Val-Comp energy and force MAE results.

TransAug-S (trained for 5 epochs) in all but one evaluation metric, particularly differentiating itself in terms of force prediction (we include the performance comparison for SPICE and reactivity splits in Table 7). We further report the performance of TransIP-S trained for 40 epochs and 80 epochs as well as TransIP-M for 60 epochs (for fair comparison to each equivariant baseline). Results with TransIP-M after 60 training epochs suggest steady improvement is likely to be observed during the remainder of the model's training epochs, which, despite limited compute, we are currently working towards. We also report the inference speed for our TransIP versions and eSEN baseline using the same hardware with 8 A100 GPUs in Table 2. For eSEN, we follow the small version indicated by Levine et al. (2025) with hyperparameters in Table 4. Both TransIP's small and medium versions are significantly faster than the eSEN baseline, while TransIP-L is slightly faster than eSEN.

| | TransIP-S | TransIP-M | TransIP-L | eSEN |
| --- | --- | --- | --- | --- |
| Approx. atoms/sec | 60,000 | 18,500 | 4,200 | 3,900 |

Table 2: Inference speed for TransIP variants and eSEN baseline.

# 7 WHAT TRANSIP LEARNS

To understand the structure of learned equivariance, we ask whether the effect of rotating different inputs can be explained by a *single* group action in the latent space; i.e., whether there exists a representation $\rho(g) : \mathbb{R}^d \to \mathbb{R}^d$ such that $f(\phi(g)(m)) \approx \rho(g) f(m)$, where $f_\theta : \mathcal{M} \to \mathbb{R}^d$ denotes the embedding network, and $g \in \mathrm{SO}(3)$ acts on a molecule $m$ via the input representation $\phi(g)$ (rotation of atomic coordinates). Because $\rho(g)$ is unknown, we compute an approximate group action $\widehat{\rho}(g) \in \mathrm{O}(d)$ by solving an orthogonal Procrustes problem on embeddings from 100 validation samples (obtained from a trained TransIP model). Writing $Z = [\, f(m_1)^\top, \ldots, f(m_n)^\top \,]$, $\quad Z_g = [\, f(\phi(g)(m_1))^\top, \ldots, f(\phi(g)(m_n))^\top \,]$, we first pool-whiten the two views (shared mean and standard deviation per channel) and then solve $\widehat{\rho}(g) = \arg\min_{Q \in \mathrm{O}(d)} \big\| \widetilde{Z}Q - \widetilde{Z}_g \big\|_F^2$, which has the closed form $\widehat{\rho}(g) = UV^\top$ for the SVD of $\widetilde{Z}^\top \widetilde{Z}_g = U\Sigma V^\top$.

In Figure 5a, we report per-molecule residuals before alignment, $\| f(m) - f(\phi(g)(m)) \|_2$, and after applying the global orthogonal map, $\| \widehat{\rho}(g) f(m) - f(\phi(g)(m)) \|_2$. A left→right drop in the distribution indicates that a single orthogonal transform explains most of the rotation-induced change in the embedding. In Figure 5b, we compare the channel-level relation by plotting a hexbin density of all pairs $(\widehat{\rho}(g) f(m))_k$, $(f(\phi(g)(m)))_k$, $k = 1, \ldots, d$, $m \in$ val. where color encodes the log count of points in each hexagonal bin. A tight diagonal concentration after the single global alignment $\widehat{\rho}(g)$ might suggest that the two views are almost identical at entrywise-level and the group action in latent space is *approximately orthogonal* and shared across different molecules.

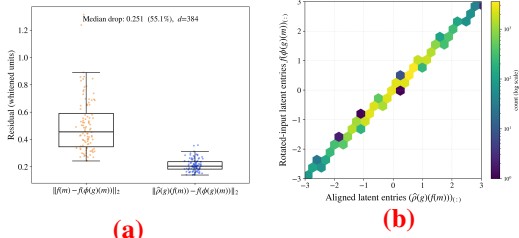

**(a)**           **(b)**

Figure 5: **Group action in the embedding space. (a)** Per-molecule residuals before alignment, $\| f(m) - f(\phi(g)(m)) \|_2$, and after applying a global orthogonal map $\widehat{\rho}(g)$ on pool-whitened latents, $\| \widehat{\rho}(g) f(m) - f(\phi(g)(m)) \|_2$. **(b)** Entrywise comparison: hexbin density of $(\widehat{\rho}(g) f(m))_{(:)}$ vs. $f(\phi(g)(m))_{(:)}$, pooled over molecules' embeddings.

**Takeaways.** Figure 5a shows that the magnitude of the rotation-induced discrepancy of different molecules drops after a single orthogonal alignment, and Figure 5b shows that the aligned channels match entrywise, concentrating along the identity. These results indicate that TransIP learns an embedding where input rotations act approximately as a shared orthogonal transformation, even though explicit equivariance was not enforced in the architecture.

# 8 CONCLUSION

In this work, we introduced TransIP for modeling interatomic potentials with a modern Transformer-based architecture and a scalable latent equivariance objective. Empirical results across a variety of chemical systems as well as model and dataset scales suggest that TransIP's latent equivariance objective enables better performance scaling than popular data augmentation-based alternatives to learning geometric equivariance. Further, we find that improvements in learning latent equivariance are strongly related to improved modeling of interatomic potentials, suggesting a complementary nature between the two prediction objectives. With sufficient compute, future work could involve studying the performance of TransIP in larger data, modeling, and runtime regimes in addition to the behavior of TransIP in a context amenable to the double-descent phenomenon (Power et al., 2022).

While equivariant models for molecular machine learning have recently gained much research interest, with the large amount of data being generated and the need for larger model sizes, it is also important that models used for interatomic potentials be highly scalable. Through our work, we have shown that the generic Transformer is capable of modeling molecules accurately but is also able to learn equivariance effectively through our novel latent objective, all while being highly scalable. By making our code openly available to the research community, we hope that our work inspires future research that explores ways to leverage the simpler and more scalable Transformer architecture to better model equivariant molecular properties through learned equivariance.

## 9 ETHICS STATEMENT

This work focuses on developing scalable machine-learned interatomic potentials (MLIPs). Our contributions are based on the principle of equivariance and do not involve sensitive personal data, human subjects, or personally identifiable information. The dataset used, OMol25, is an open-source quantum chemistry benchmark for research use. The potential broader impacts of our method are that it can accelerate research in drug discovery and material sciences. By making MLIP models cheaper to train, we make molecular modeling more accessible to the broader research community. However, misuse in safety-critical applications like drug discovery could lead to adverse outcomes. No dual-use is identified beyond the general risks of over-reliance on approximate ML models in scientific workflows.

## 10 REPRODUCIBILITY STATEMENT

**Dataset**: All experiments were conducted on the Open Molecules 2025 (OMol25) dataset, which is publicly available and documented (Levine et al., 2025). We follow its official training and validation splits (4M train dataset, 2M out-of-distribution validation dataset).

**Architectures and hyperparameters**: We include detailed architectural configurations (e.g., model sizes, layers, hidden dimensions, attention heads) in Appendix A.

**Evaluation**: We report standardized OMol25 metrics: Force MAE, Force cosine similarity, Energy per atom MAE, and Total Energy MAE.

**Code and models**: We build our implementation on the FAIRCHEM framework for standardized MLIP training and evaluation. However, we also plan to release our code as an open-source repository upon acceptance.

**Scaling experiments**: We perform scaling experiments with varying model and dataset sizes, with corresponding results presented in Figures 2, 3, and 4.

**Compute resources**: Experiments were run on a limited number of NVIDIA A100 80GB GPUs. Small-scale experiments used 8 GPUs for 5 epochs, while extended runs used 64 GPUs for up to 80 epochs (currently in progress).

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

# A    IMPLEMENTATION DETAILS

## A.1    MODEL ARCHITECTURE

Table 3 provides the complete architectural specifications for TransIP's model versions, as well as eSEN hyperparameters for the inference test in Table 2. For eSEN, we follow the small version reported by Levine et al. (2025).

Table 3: TransIP model configurations. All versions share the same embedding method and activation functions.

| Configuration | Small (S) | Medium (M) | Large (L) |
|---|---|---|---|
| Hidden dimension (d) | 384 | 768 | 1024 |
| Number of layers (L) | 8 | 12 | 24 |
| Number of heads | 6 | 12 | 16 |
| Total parameters | 14M | 85M | 302M |
| *Shared configurations:* | | | |
| Coordinate embedding | | MLP | |
| Activation function | | GELU | |
| Context length | | 1024 | |
| Projection dropout | | 0.01 | |
| Attention dropout | | 0.0 | |
| *Transformation network $\mathcal{T}_\tau$:* | | | |
| Number of layers | | 2 | |
| Hidden dimension | | $2 \times d$ | |
| Activation | | GELU | |

Table 4: eSEN hyperparameters for inference test in Table 2.

| Configuration | Value |
|---|---|
| sphere_channels | 128 |
| lmax | 2 |
| mmax | 2 |
| edge_channels | 128 |
| distance_function | gaussian |
| num_distance_basis | 64 |
| num_layers | 4 |
| hidden_channels | 128 |
| max_neighbors | 30 |
| cutoff_radius | 6 |
| normalization_type | rms_norm_sh |
| activation_type | gate |
| ff_type | spectral |

## A.2    TRAINING HYPERPARAMETERS

Table 5 provides TransIP's optimal hyperparameters.

## A.3    DATA PROCESSING AND AUGMENTATION

TransIP processes molecular data with the following pipeline:

- **Coordinate centering**: Atomic coordinates are centered by subtracting the center of mass:
  $\mathbf{r}_i \leftarrow \mathbf{r}_i - \frac{1}{|m|} \sum_j \mathbf{r}_j$

Table 5: Training hyperparameters used for all TransIP experiments.

| Hyperparameter | Value |
|---|---|
| *Optimization:* | |
| Optimizer | AdamW |
| Learning rate | $5 \times 10^{-4}$ |
| Weight decay | $1 \times 10^{-3}$ |
| Gradient clip norm | 200 |
| *Learning rate schedule:* | |
| Scheduler type | Cosine |
| Warmup fraction | 0.01 |
| Min LR factor | 0.01 |
| *Loss weights:* | |
| Energy ($\lambda_E$) | 5 |
| Forces ($\lambda_F$) | 15 |
| Equivariance ($\lambda_{leq}$) | 5 (selected from $\{1, 5, 10, 100\}$) |

- **Equivariance pairs**: For training with learned equivariance, we create pairs $(m, \phi(g)(m))$ where $g$ is sampled uniformly from $SO(3)$ per molecule.

## A.4 EVALUATION METRICS

We evaluate model performance using the following metrics:

**Force Mean Absolute Error (MAE):**

$$\text{Force MAE} = \frac{1}{3|m|} \sum_{i=1}^{N} \sum_{\alpha \in \{x,y,z\}} |\mathbf{F}_{i,\alpha} - \mathbf{F}_{i,\alpha}^*| \quad (\text{meV/Å}) \tag{8}$$

**Force Cosine Similarity:**

$$\text{Force CosSim} = \frac{1}{|m|} \sum_{i=1}^{|m|} \frac{\mathbf{F}_i \cdot \mathbf{F}_i^*}{\|\mathbf{F}_i\| \|\mathbf{F}_i^*\|} \tag{9}$$

**Energy per Atom MAE:**

$$\text{Energy/atom MAE} = \frac{1}{|m|} |E - E^*| \quad (\text{meV/atom}) \tag{10}$$

**Total Energy MAE:**

$$\text{Total Energy MAE} = |E - E^*| \quad (\text{meV}) \tag{11}$$

where $\mathbf{F}$ and $E$ denote predicted forces and energies, $\mathbf{F}^*$ and $E^*$ are ground truth values, and $|m|$ is the total number of atoms. For energies, we use referenced targets following Levine et al. (2025).

## A.5 COMPUTATIONAL RESOURCES

- 5-epoch experiments: 8 NVIDIA 80GB GPUs
- 80-epoch experiments: 64 NVIDIA 80GB GPUs

## A.6 VALIDATION SPLITS

For 5-epoch runs, we evaluate on domain-specific validation subsets sampled from the OMol25 validation (Val-Comp) dataset:

- Metal complexes: 20,000 samples

- Electrolytes: 20,000 samples
- Biomolecules: 20,000 samples
- SPICE: 9,630 samples (complete subset)
- Neutral organics: 20,000 samples (including ANI2x, OrbNet-Denali, GEOM, Trans1x, RGD)
- Reactivity: 20,000 samples
- Full validation set: 20,000 samples.

We use the full (2M) Val Comp dataset to evaluate TransIP and TransAug in Table 1.

# B  ADDITIONAL RESULTS

## B.1  OPEN CATALYST BENCHMARK

We also evaluate our method on the Open Catalyst 2020 (OC20) benchmark (Chanussot et al., 2021), a large-scale dataset for modeling catalyst-adsorbate interactions. We train on the 2M subset from the Structure-to-Energy-and-Forces (S2EF) task, and for validation, we selected $20,000$ samples from each validation split: val_id (in-distribution) and val_ood (out-of-distribution). We use the small version of TransIP and TransAug with the same hyperparameters in Tables 3 and 5, trained for 30000 steps. Our results are presented in Table 6. Our results show that TransIP consistently outperforms TransAug on energy metrics on both in-distribution and out-of-distribution splits, and matches TransAug on force MAE.

| Model | val_id | | val_ood | |
|---|---|---|---|---|
| | Energy ↓ | Forces ↓ | Energy ↓ | Forces ↓ |
| TransAug-S | 56 | 82 | 72 | 95 |
| TransIP-S | 45 | 82 | 55 | 95 |

Table 6: OC20 S2EF energy and force MAE on val_id and val_ood splits.

## B.2  OMOL25 SPLITS

In this section, we include additional dataset scaling results on OMol25 splits for TransIP and TransAug.

| Model | Epochs | SPICE | | Reactivity | |
|---|---|---|---|---|---|
| | | Energy ↓ | Forces ↓ | Energy ↓ | Forces ↓ |
| TransAug-S | 5 | 11.5 | 151.3 | 23.0 | 179.7 |
| TransIP-S | 5 | 8.7 | 121.8 | 17.8 | 136.4 |

Table 7: Val-Comp energy and force MAE for SPICE and Reactivity splits.

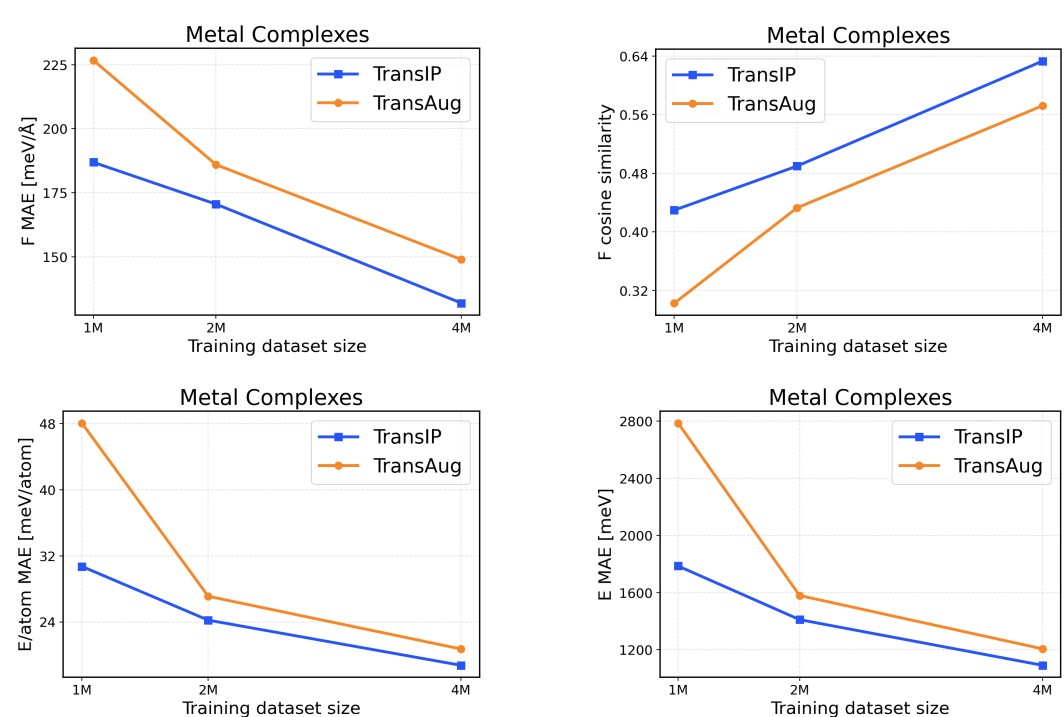

Figure 6: Metal Complexes scaling across training dataset sizes (1M / 2M / 4M). The top row presents force metrics, while the bottom row displays energy metrics.

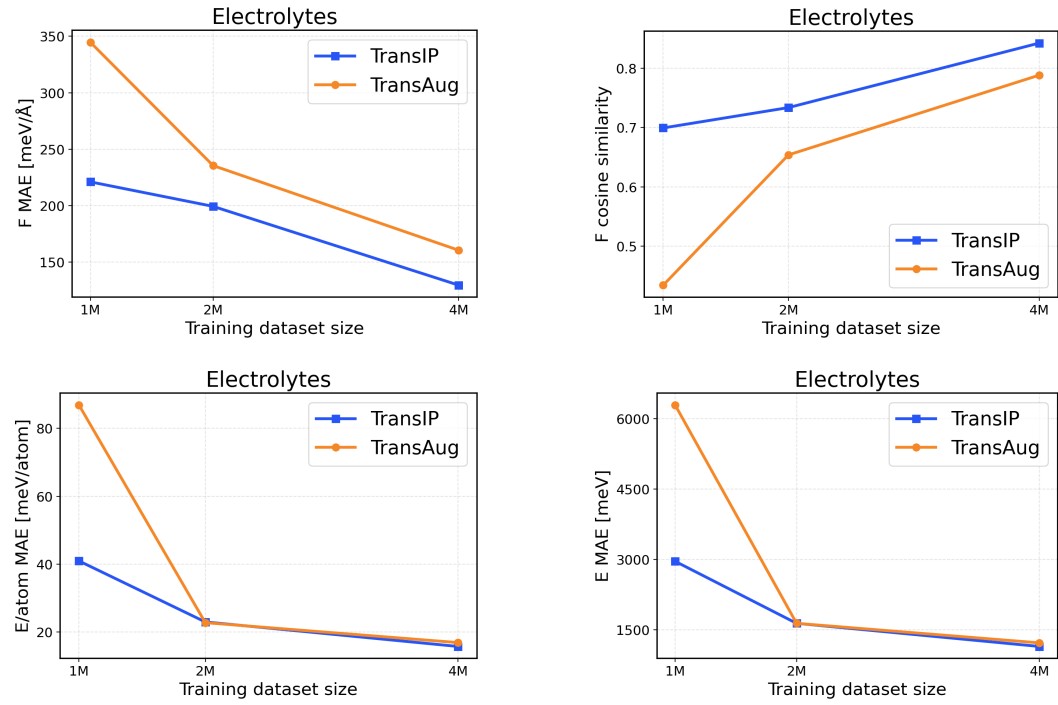

Figure 7: Electrolytes scaling across training dataset sizes (1M / 2M / 4M). The top row presents force metrics, while the bottom row displays energy metrics.

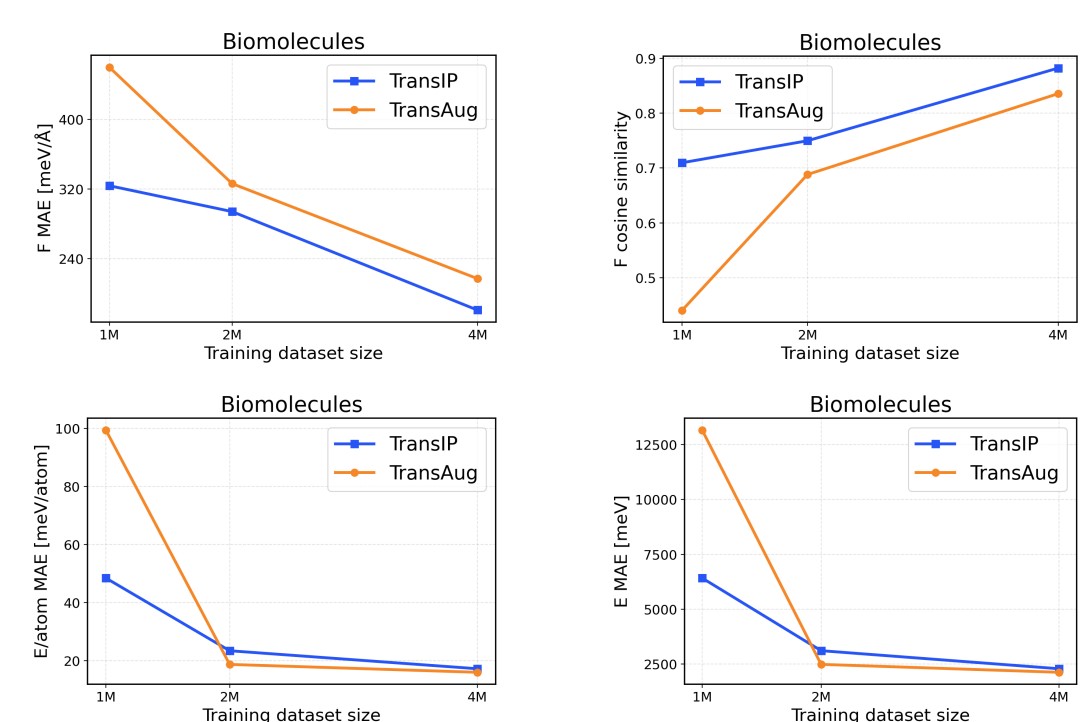

Figure 8: Biomolecules scaling across training dataset sizes (1M / 2M / 4M). The top row presents force metrics, while the bottom row displays energy metrics.

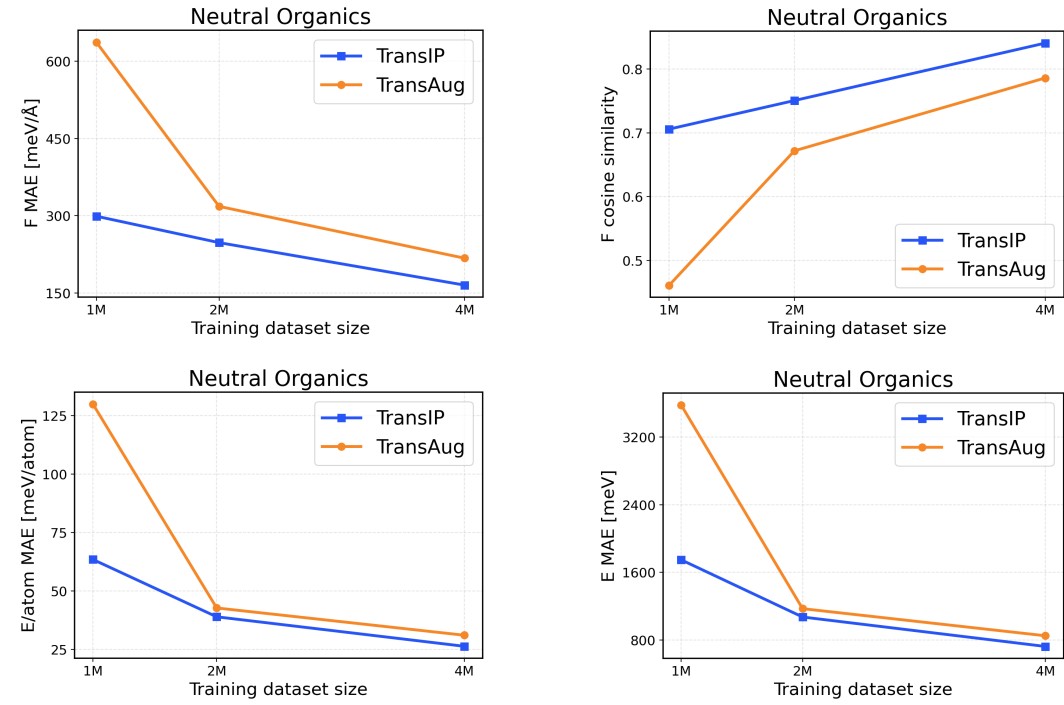

Figure 9: Neutral Organics scaling across training dataset sizes (1M / 2M / 4M). The top row presents force metrics, while the bottom row displays energy metrics.

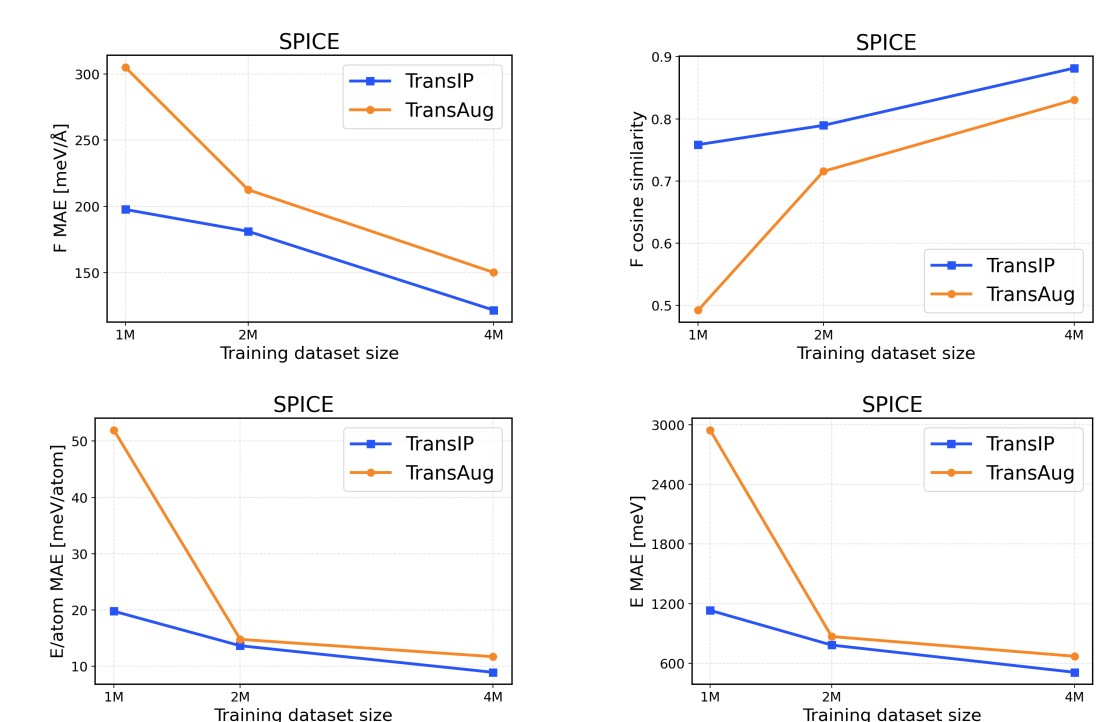

Figure 10: SPICE scaling across training dataset sizes (1M / 2M / 4M). The top row presents force metrics, while the bottom row displays energy metrics.

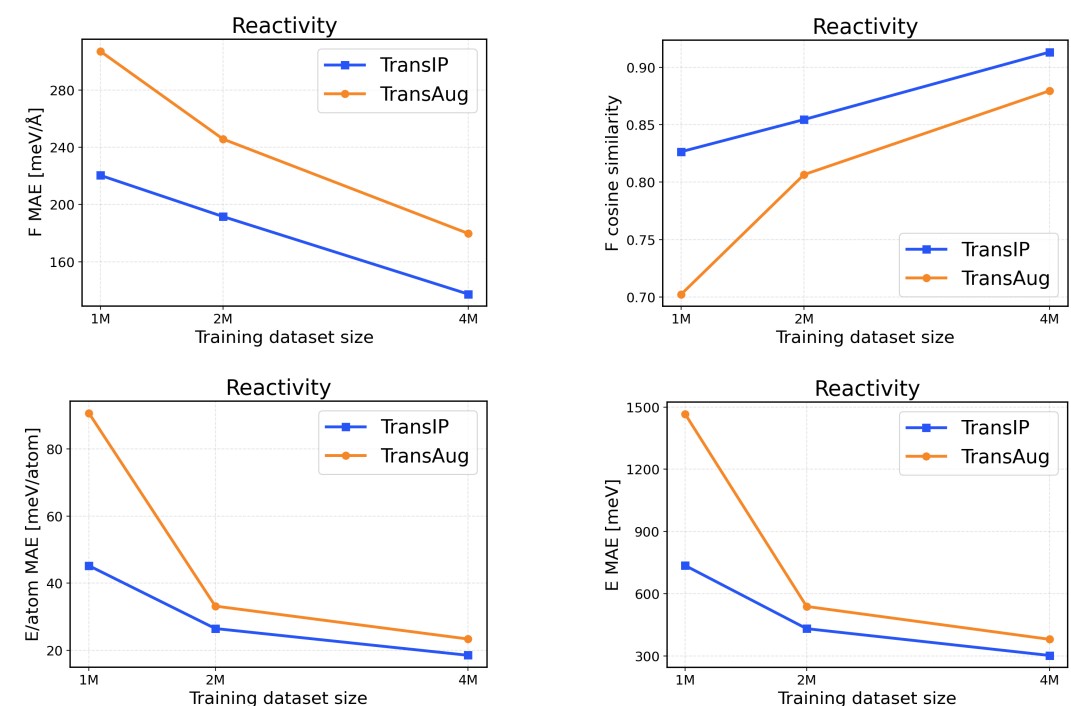

Figure 11: Reactivity scaling across training dataset sizes (1M / 2M / 4M). The top row presents force metrics, while the bottom row displays energy metrics.

