# OpenReview forum: "Learning Inter-Atomic Potentials without Explicit Equivariance"
_ICLR.cc/2026/Conference — ICLR 2026 Conference Withdrawn Submission_

### Official Review · Reviewer_RAy9 · 2025-10-15

**Soundness:** 3
**Presentation:** 3
**Contribution:** 3
**Rating:** 4
**Confidence:** 5

**Summary:**

While the proposed TransIP achieves faster inference and improved scalability, the experimental comparison to equivariant baselines is insufficient to substantiate the claimed “comparable accuracy.”

**Strengths:**

The idea of learning latent equivariance through a contrastive objective is conceptually interesting and practically promising.

**Weaknesses:**

While the proposed TransIP achieves faster inference and improved scalability, the experimental comparison to equivariant baselines is insufficient to substantiate the claimed “comparable accuracy.” Specifically, Table 1 reports results only against eSCN and GemNet-OC, but the configuration details (number of parameters, training epochs, and compute budgets) are not provided for these models.
Without matching model size or FLOPs, the efficiency-accuracy trade-off is difficult to interpret.

Although TransIPreaches ∼0.10 eV Å⁻¹ in force MAE, the best eSCN models achieve ∼0.01 eV Å⁻¹. This difference of nearly one order of magnitude cannot be considered “comparable.”

The study omits several strong and more recent baselines that represent the current frontier in equivariant MLIPs, such as SO3Krates, ViSNet, MACE.

Figure 4 qualitatively shows “faster inference,” but numerical throughput (atoms/s) is not tabulated, and wall-clock time per step or per molecule is not reported.

**Questions:**

See Section Weaknesses.

---

> ### Author Response · Authors · 2025-11-22
>
> We thank the reviewer for their time and all the comments, which have helped us improve our paper and indicate our contribution.
>
> > "While the proposed TransIP achieves faster inference and improved scalability, the experimental comparison to equivariant baselines is insufficient to substantiate the claimed “comparable accuracy.” Specifically, Table 1 reports results only against eSCN and GemNet-OC, but the configuration details (number of parameters, training epochs, and compute budgets) are not provided for these models. Without matching model size or FLOPs, the efficiency-accuracy trade-off is difficult to interpret.  Although TransIPreaches ∼0.10 eV Å⁻¹ in force MAE, the best eSCN models achieve ∼0.01 eV Å⁻¹. This difference of nearly one order of magnitude cannot be considered “comparable.”
>
> **Author Response:** We thank the reviewer for this point. We would like to point out to the reviewer that the baselines are trained for 80 epochs as indicated by [1]. Due to the limited compute, we are able to train our medium model now for 60 epochs. We have also updated the results in the revised manuscript, where TransIP comes closer to the eSEN baseline. That is, for forces, TransIP-M achieves 35.4 meV/Å vs. 13 meV/Å compared to eSEN. We will also report the medium model’s 80-epoch results in the final version of the paper.   However, we would like to point out to the reviewer that the medium model is still significantly faster than the eSEN baseline, where eSEN has inference speed comparable to the large version of TransIP, as indicated below.
>
> > "The study omits several strong and more recent baselines that represent the current frontier in equivariant MLIPs, such as SO3Krates, ViSNet, MACE."
>
> **Author Response:** We thank the reviewer for this point. We would like to point out to the reviewer that the eSEN and GemNet-OC baselines are two of the state-of-the-art baselines on this benchmark, and they already outperform MACE, Equifromer, Orb, DimNet, SpinConv, and many other equivariant models (see [1, 2, 3]).
>
>
> > "Figure 4 qualitatively shows “faster inference,” but numerical throughput (atoms/s) is not tabulated, and wall-clock time per step or per molecule is not reported."
>
> **Author Response:** We thank the reviewer for this point. We have now included inference speed for our models and eSEN baseline using the same hardware with 8 A100 GPUs in Table 2 in the revised version of the manuscript. Both TransIP’s small and medium versions are significantly faster than the eSEN baseline, while TransIP-L is slightly faster than eSEN.
>
>
> We sincerely hope that we have addressed the concerns of the reviewer in the revised manuscript of the paper, and would kindly ask the reviewer to update their score accordingly.
>
> References:
>
> 1. The Open Molecules 2025 (OMol25) Dataset, Evaluations, and Models. Levine et al., 2025.
> 2. Learning Smooth and Expressive Interatomic Potentials for Physical Property Prediction. Fu et al., ICML 2025..
> 3. GemNet-OC: Developing Graph Neural Networks for Large and Diverse Molecular Simulation Datasets. Gasteiger et al., TMLR 2022.

---

> > ### Comment · Reviewer_RAy9 · 2025-11-24
> >
> > Thank you for your reply. However, the response did not address my concerns, so I will be keeping my original rating.

---

> > > ### Author Response · Authors · 2025-11-24
> > >
> > > We thank the reviewer again for their time and for engaging with us in the discussion. We kindly ask the reviewer to check the updated version of the manuscript. If there are additional issues that we may have overlooked or if anything remains unclear, we would greatly appreciate further clarification from the reviewer.

---

### Official Review · Reviewer_RoKz · 2025-10-30

**Soundness:** 4
**Presentation:** 3
**Contribution:** 3
**Rating:** 4
**Confidence:** 4

**Summary:**

This paper presents TransIP, a Transformer-based framework for learning interatomic potentials without explicit SO(3)-equivariant architectures. The authors propose a training scheme that achieves learned equivariance in embedding space through a contrastive latent objective. Instead of using tensor products or spherical harmonics, TransIP encourages the model to learn rotational consistency directly from data, improving flexibility and scalability compared to conventional equivariant MLIPs.

The method shows comparable performance to state-of-the-art equivariant models and notably outperforms data augmentation baselines, particularly in low-data regimes. The study positions TransIP as a bridge between fully constrained equivariant networks and unconstrained Transformers.

Overall, this is a well-motivated and clearly scoped investigation into learned symmetry for molecular systems, though additional probing of the latent representation and model behavior under symmetry operations would substantially strengthen the work.

**Strengths:**

* Introduces a clear, architecture-agnostic approach for encouraging SO(3) equivariance in a Transformer backbone without explicit geometric layers.
* Demonstrates strong empirical performance relative to augmentation-based baselines, validating the utility of learned equivariance.
* Employs rotary position embeddings (RoPE) to inject coordinate dependence without fixed distance cutoffs
* Provides a scalable alternative to equivariant message-passing networks, highlighting potential for deployment on large molecular datasets.
* The experiments on scaling (dataset and model size) are thoughtfully structured and well contextualized.

**Weaknesses:**

* Enforcing equivariance is only explored on the final pooled molecular embedding, not at the per-atom or intermediate-layer level. As a result, local geometric information is only indirectly constrained and cannot be explicitly evaluated.
* The contrastive loss may over-regularize or under-constrain the latent space; a discussion or ablation on this sensitivity is missing.
* The current benchmark results are fine, but more investigation on the latent space would really enhance the paper's contributions.
* Evaluation tables should follow community standards (e.g., reporting energies in meV and forces in meV/Å) and include units on all axes (e.g., Fig. 3).
* The literature review could better acknowledge earlier works using spherical harmonics for convolutional equivariance prior to Gasteiger et al. (2020) and Klicpera et al. (2021), along the lines of 3D steerable CNNs, Cormorant, Tensor Field Networks, etc.

**Questions:**

1. **Latent structure and decomposition:**
   Can the authors measure how much of the latent representation becomes invariant vs. equivariant under the learned transformation? Since the final target (energy) is invariant and forces are derived via gradients, it would be insightful to quantify what fraction of latent dimensions encode rotational behavior.
2. **Layerwise analysis:**
  It would be interesting to evaluate how the latents per layer. This would help illustrate whether early layers learn geometry or whether equivariance only appears late.
3. **Intermediate supervision:**
  Have you tested applying the contrastive loss on intermediate latent representations or per-atom embeddings? Does this improve or degrade performance, and how sensitive is it to the strength of the loss? (It would be especially interesting to see whether standard nonlinearities—most of which break equivariance—prevent stable intermediate supervision.)
4. **Per-atom vs. global equivariance:**
  Currently, the loss is applied only to the final aggregated molecular embedding. In principle, a per-atom latent equivariance loss could improve local geometric fidelity. Could you comment on whether this was attempted or ruled out due to stability or computational cost?
5. **Translation symmetry and RoPE:**
Can you elaborate more in the text on why you chose the RoPE embeddings. Currently one needs to read through RoPE to get this and it would be helpful for the read to not have to do that -- it breaks focus while reading. Additionally, please comment on how RoPE does / does not handles translation symmetry. Since the model centers coordinates by the center of mass, how does it behave for systems lacking a natural origin (e.g., extended periodic systems)?
5. **Metric conventions:**
For consistency and clarity, please provide units in Fig. 3 and report energies in meV and forces in meV/Å in Table 1.

---

> ### Author Response · Authors · 2025-11-22
>
> We thank the reviewer for their time and all the comments, which have helped us improve our paper and indicate our contribution.
>
> > "Enforcing equivariance is only explored on the final pooled molecular embedding, not at the per-atom or intermediate-layer level."
>
> **Author Response:** We thank the reviewer for this point. Given a general class of functions $f$ with embedding space in $R^d$, we want the final function to be equivariant, so we optimize the backbone with the contrastive loss. Applying individual loss per layer might have additional complexity, see for example [1]. Our loss is currently applied as a transformation for input $x$ where a rotation affects the molecule’s atoms together, so we rotate atoms of the same molecule with the same rotations. Applying different rotations per atom would not be considered a valid $SO(3)$ transformation.
>
>
> > "The contrastive loss may over-regularize or under-constrain the latent space; a discussion or ablation on this sensitivity is missing. The current benchmark results are fine, but more investigation on the latent space would really enhance the paper's contributions."
>
> **Author Response:** We thank the reviewer for these points. In the revised manuscript, we have included an ablation study on the effect of the contrastive objective on the latent space. In this experiment, we ask whether the effect of rotating different inputs can be explained by a single group action in the latent space, i.e., whether there exists a representation $\rho(g):\mathbb{R}^d\to\mathbb{R}^d$ such that
> $f\bigl(\phi(g)(m)\bigr)\approx\rho(g)f(m)$,
> where $f_\theta:\mathcal{M}\to\mathbb{R}^d$ denotes the embedding network, and $g\in\mathrm{SO}(3)$ acts on a molecule $m$ via the input representation $\phi(g)$ (rotation of atomic coordinates). Because $\rho(g)$ is unknown, we compute an approximate group action $\widehat{\rho}(g)\in\mathrm{O}(d)$ by solving an orthogonal Procrustes problem on embeddings from validation samples obtained from a trained TransIP model. More details are explained in Section 7 in the revised paper.
> In Figure 5.a, we report per-molecule residuals before alignment, $\|\|f(m)-f(\phi(g)(m))\|\|_2$, and after applying the global orthogonal map, $\|\|\widehat{\rho}(g)f(m)-f(\phi(g)(m))\|\|_2$.
> Our results show that the magnitude of the rotation-induced discrepancy of different molecules drops after a single orthogonal alignment A drop in the distribution indicates that a single orthogonal transform explains most of the rotation-induced change in the embedding.
>
> In Figure 5.b, we also compare the channel-level relation by plotting a hexbin density of all pairs
> $(\widehat{\rho}(g)f(m))_k,\qquad (f(\phi(g)(m)))_k,\qquad k=1,\dots,d,\; m\in\text{val}$
>
> Here, we noticed a tight diagonal concentration after the single global alignment $\widehat{\rho}(g)$. These results suggest that the two views are almost identical at entrywise-level and the group action in latent space is approximately orthogonal and shared across different molecules.
>
> > "Evaluation tables should follow community standards (e.g., reporting energies in meV and forces in meV/Å) and include units on all axes (e.g., Fig. 3)."
>
> **Author Response:** We thank the reviewer for this point. In the revised manuscript, we have updated tables now with meV and meV/Å units for energies and forces, respectively, and include them in all figures.
>
> > "The literature review could better acknowledge earlier works using spherical harmonics for convolutional equivariance prior to Gasteiger et al. (2020) and Klicpera et al. (2021), along the lines of 3D steerable CNNs, Cormorant, Tensor Field Networks, etc."
>
> **Author Response:** We thank the reviewer for this point. We have now included steerable CNNs, Cormorant, and Tensor Field Networks in the related work section.
>
> > Translation symmetry and RoPE.
>
> We thank the reviewer for this point. We would like to point out to the reviewer that rotary position embedding (RoPE) is one of the current common encoding methods with which to inject relative positional information into the attention mechanism. Also, RoPE itself only depends on atom indices within the sequence (not on absolute coordinates), so a global translation of all atom positions leaves the RoPE factors unchanged, which differs from the translation invariance of atomic coordinates.
>
> We sincerely hope that we have addressed the concerns of the reviewer in the revised manuscript of the paper, and would kindly ask the reviewer to update their score accordingly.
>
>
> References:
>
> 1. Learning Layer-wise Equivariances Automatically using Gradients. van der Ouderaa et al., NeurIPS 2023.

---

> > ### Author Response · Authors · 2025-11-26
> > **Follow Up**
> >
> > Dear reviewer,
> >
> >
> > Thank you once again for your time and the valuable feedback you have provided. We have made our best effort to answer your questions and address your comments in the updated version of the manuscript. We kindly ask the reviewer to consider increasing their score. We are also happy to engage with the reviewer if they have any further questions.

---

> > > ### Author Response · Authors · 2025-11-28
> > >
> > > Thank you for increasing your score to 6 on Nov 26.

---

### Official Review · Reviewer_emGi · 2025-11-01

**Soundness:** 3
**Presentation:** 3
**Contribution:** 2
**Rating:** 4
**Confidence:** 4

**Summary:**

The paper introduces TransIP, which steers a standard Transformer toward SO(3) equivariance via a latent equivariance (contrastive) objective in feature space, avoiding hard-wired equivariant layers. The model adds a loss term that aligns the features of a molecule with its rotated embeddings via a learnable transform  T_τ. On OMol25 dataset, TransIP outperforms rotation-augmentation baselines and achieves performance comparable to equivariant MLIP baselines.

**Strengths:**

A friendly way to add equivalence. The core idea of this paper is clear. The latent equivariance loss is architecture-agnostic and straightforward to implement, allowing the backbone to remain unconstrained while satisfying equivariance by avoiding the use of spherical tensors.

**Weaknesses:**

Focus on molecules only. Experimental results only target OMol25. SPICE and its SOTA benchmarks are not discussed.
Ablations on  T_τ are limited. The core contribution is an implicit equivariance module(a learnable transform). However, the evaluation does not display the effect of the learned equivariance itself from other invariant MLIPs. The authors should add the implicit-equivariance module to other invariant backbones and compare with the invariant-only MLIPs.
The results are not good enough. Based on the results in Table 1, TransIP does not outperform the competing models. In fact, its accuracy is generally lower than that of eSEN and GemNet.

**Questions:**

Please refer to the weaknesses.

---

> ### Author Response · Authors · 2025-11-22
>
> We thank the reviewer for their time and all the comments, which have helped us improve our paper and indicate our contribution.
>
> > "Focus on molecules only. Experimental results only target OMol25. SPICE and its SOTA benchmarks are not discussed."
>
> **Author Response:** We thank the reviewer for this point. We would like to point out that OMol25 [1] is a diverse molecular dataset that covers 83 atomic elements and six primary categories: metal complexes, electrolytes, biomolecules, SPICE, neutral organics, and reactivity. We now explain this at the beginning of Section 5 in the revised version. We also include the dataset scaling results (1M, 2M, and 4M) comparing TransIP and TransAug for each category in Appendix B.2. Additionally, we report the results for SPICE and reactivity in Table 7 in the revised version:
>
> | Model    | SPICE Energy ↓ | SPICE Forces ↓ | Reactivity Energy ↓ | Reactivity Forces ↓ |
> |-----------|:-------------:|---------------:|--------------------:|---------------------:|
> | TransAug-S    | 11.5           | 151.3          | 23.0                | 179.7                |
> | TransIP-S    | 8.7            | 121.8          | 17.8                | 136.4                |
>
>
> We noticed these results are consistent with other molecule splits in the paper, where TransIP significantly outputs TransAug on both energy and force metrics.
>
> **Open Catalyst benchmark:** We have now included new evaluations on the Open Catalyst 2020 (OC20) benchmark [2].  We train on the 2M subset from the Structure-to-Energy-and-Forces (S2EF) task, and for validation, we selected $20,000$ samples from each validation split: `val_id` (in-distribution) and `val_ood` (out-of-distribution).  We use the small version of TransIP and TransAug with the same hyperparameters in Tables 3 and 5 in the updated version of the paper, trained for 30,000 steps. We include the results in Table 6, which shows that TransIP outperforms TransAug on energy metrics on both in-distribution and out-of-distribution splits, and matches TransAug on force metrics.
>
> | Model      | `val_id` Energy ↓ | `val_id` Forces ↓ | `val_ood` Energy ↓ | `val_ood` Forces ↓ |
> |-----------|-------------------:|-------------------:|--------------------:|---------------------:|
> | TransAug-S | 56                | 82                | 72                 | 95                  |
> | TransIP-S  | 45                | 82                | 55                 | 95                  |
>
> > "The core contribution is an implicit equivariance module(a learnable transform). However, the evaluation does not display the effect of the learned equivariance itself from other invariant MLIPs. The authors should add the implicit-equivariance module to other invariant backbones and compare with the invariant-only MLIPs."
>
> **Author Response:** We thank the reviewer for this point. We would like to point out to the reviewer that our latent equivariance loss can be applied to any unconstrained architecture in the latent representation $z=f(x) \in R^d$. However, if the function $f$ is strictly invariant, i.e. $f(gx)=f(x)$ for all $g \in SO(3)$, then enforcing a relation of the form  $ρ(g) f(x)≈f(gx)$  is trivial with the optimal transformation in latent space will be the identity map  $\rho(g) = I$. In this case, we can relax the strict invariance model to an approximate invariance backbone. For example, we could design an additional embedding from the 3D coordinates that is combined with invariant features, which we leave for future work.
>
> > "The results are not good enough. Based on the results in Table 1, TransIP does not outperform the competing models. In fact, its accuracy is generally lower than that of eSEN and GemNet."
>
> **Author Response:** We thank the reviewer for this point. We would like to point out to the reviewer that the baselines are trained for 80 epochs as indicated by [1]. Due to the limited compute, we are able to train our medium model now for 60 epochs. We have also updated the results in the revised manuscript, where TransIP comes closer to the eSEN baseline. That is, for forces, TransIP-M achieves 35.4 meV/Å vs. 13 meV/Å compared to eSEN. We will also report the 80-epoch results for medium in the final version of the paper. Additionally, the TransIP medium version is still significantly faster than the eSEN baseline, while TransIP-L is slightly faster than eSEN, as indicated in Table 2 in the revised version of the paper.
>
> We sincerely hope that we have addressed the concerns of the reviewer in the revised manuscript of the paper, and would kindly ask the reviewer to update their score accordingly.
>
>
> References:
>
> 1. The Open Molecules 2025 (OMol25) Dataset, Evaluations, and Models. Levine et al., 2025.
> 2. The Open Catalyst 2020 (OC20) Dataset and Community Challenges. Chanussot et al., ACS 2021.

---

> > ### Author Response · Authors · 2025-11-26
> > **Follow Up**
> >
> > Dear reviewer,
> >
> >
> > Thank you once again for your time and the valuable feedback you have provided. We have made our best effort to answer your questions and address your comments in the updated version of the manuscript. We kindly ask the reviewer to consider increasing their score. We are also happy to engage with the reviewer if they have any further questions.

---

### Official Review · Reviewer_NW4M · 2025-11-01

**Soundness:** 2
**Presentation:** 3
**Contribution:** 2
**Rating:** 4
**Confidence:** 2

**Summary:**

This paper presents a Transformer-based interatomic potential prediction model called TransIP. The core innovation lies in learning SO(3) equivariance implicitly through a contrastive learning objective in the embedding space, without relying on explicit equivariant architectures. The authors validate the method on the large molecular dataset OMol25, showing that it outperforms traditional data augmentation methods both in limited data and large-scale training scenarios.

**Strengths:**

1.The MLIP training pipeline and architecture-agnostic contrastive loss function proposed in the paper are easy to follow.
2.TransIP shows a significant improvement in force prediction compared to traditional data augmentation methods.

**Weaknesses:**

1.TransIP does not perform well. The comparison with eSEN and GemNet-OC in Table 1 shows that it does not have an advantage in terms of performance.
2.The evaluation is only conducted on the OMol25 dataset and has not been extended to other datasets (e.g., SPICE, MPTrj, OMat24), failing to demonstrate the generalizability of the method.
3.The backbone uses only a Transformer-based architecture, and the effectiveness of the method on other invariant MLIPs has not been explored.

**Questions:**

1.Regarding the dataset: OMol25 is an inorganic molecular dataset, but it’s also important to verify whether TransIP delivers the same results on other types of datasets (e.g., Materials, Catalysis). Could you compare the performance of TranAug and TransIP on other datasets, such as MPTrj, OMat24, or OC20?
2.Regarding the backbone architecture: The current backbone is based on a Transformer, but it’s crucial to assess the method’s effectiveness with other architectures as well. Could you provide results for the architecture-agnostic contrastive loss function when applied to other invariant MLIPs?

---

> ### Author Response · Authors · 2025-11-22
>
> We thank the reviewer for their time and all the comments, which have helped us improve our paper and indicate our contribution.
>
> > "TransIP does not perform well. The comparison with eSEN and GemNet-OC in Table 1 shows that it does not have an advantage in terms of performance."
>
> **Author Response:** We thank the reviewer for this point. We would like to point out to the reviewer that the baselines are trained for 80 epochs as indicated by [1]. Due to the limited compute, we are able to train our medium model now for 60 epochs. We have also updated the results in the revised manuscript, where TransIP comes closer to the eSEN baseline. That is, for forces, TransIP-M achieves 35.4 meV/Å vs. 13 meV/Å compared to eSEN. We will also report the 80-epoch results for medium in the final version of the paper. Additionally, the TransIP medium version is still significantly faster than the eSEN baseline, while TransIP-L is slightly faster than eSEN, as indicated in Table 2 in the revised version of the paper.
>
> > "The evaluation is only conducted on the OMol25 dataset and has not been extended to other datasets (e.g., SPICE, MPTrj, OMat24), failing to demonstrate the generalizability of the method."
>
>
> **Author Response:** We thank the reviewer for this point. We would like to point out that OMol25 [1] is a diverse molecular dataset that covers 83 atomic elements and six primary categories: metal complexes, electrolytes, biomolecules, SPICE, neutral organics, and reactivity. We now explain this at the beginning of Section 5 in the revised version. We also include the dataset scaling results (1M, 2M, and 4M) comparing TransIP and TransAug for each category in Appendix B.2 Additionally,  we report the results for SPICE and reactivity in Table 7 in the revised version:
>
> | Model    | SPICE Energy ↓ | SPICE Forces ↓ | Reactivity Energy ↓ | Reactivity Forces ↓ |
> |-----------|:-------------:|---------------:|--------------------:|---------------------:|
> | TransAug-S    | 11.5           | 151.3          | 23.0                | 179.7                |
> | TransIP-S    | 8.7            | 121.8          | 17.8                | 136.4                |
>
>
> We noticed these results are consistent with other molecule splits in the paper, where TransIP significantly outputs TransAug on both energy and force metrics.
>
> **Open Catalyst benchmark:** We have now included new evaluations on the Open Catalyst 2020 (OC20) benchmark [2].  We train on the 2M subset from the Structure-to-Energy-and-Forces (S2EF) task, and for validation, we selected $20,000$ samples from each validation split: `val_id` (in-distribution) and `val_ood`  (out-of-distribution).  We use the small version of TransIP and TransAug with the same hyperparameters in Tables 3 and 5 in the updated version of the paper, trained for 30,000 steps. We include the results in Table 6, which shows that TransIP outperforms TransAug on energy metrics on both in-distribution and out-of-distribution splits, and matches TransAug on force MAE.
>
> | Model      | `val_id` Energy ↓ | `val_id` Forces ↓ | `val_ood` Energy ↓ | `val_ood` Forces ↓ |
> |-----------|-------------------:|-------------------:|--------------------:|---------------------:|
> | TransAug-S | 56                | 82                | 72                 | 95                  |
> | TransIP-S  | 45                | 82                | 55                 | 95                  |
>
>
> > "The backbone uses only a Transformer-based architecture, and the effectiveness of the method on other invariant MLIPs has not been explored."
>
> **Author Response:** We thank the reviewer for this point. We would like to point out to the reviewer that our latent equivariance loss can be applied to any unconstrained architecture in the latent representation $z=f(x) \in R^d$. However, if the function $f$ is strictly invariant, i.e. $f(gx)=f(x)$ for all $g \in SO(3)$, then enforcing a relation of the form  $ρ(g) f(x)≈f(gx)$  is trivial with the optimal transformation in latent space will be the identity map $\rho(g) = I$. In this case, we can relax the strict invariance model to an approximate invariance backbone. For example, we could design an additional embedding from the 3D coordinates that is combined with invariant features, which we leave for future work.
>
> We sincerely hope that we have addressed the concerns of the reviewer in the revised manuscript of the paper, and would kindly ask the reviewer to update their score accordingly.
>
> References:
>
> 1. The Open Molecules 2025 (OMol25) Dataset, Evaluations, and Models. Levine et al., 2025.
> 2. The Open Catalyst 2020 (OC20) Dataset and Community Challenges. Chanussot et al., ACS 2021.

---

> > ### Author Response · Authors · 2025-11-26
> > **Follow Up**
> >
> > Dear reviewer,
> >
> >
> > Thank you once again for your time and the valuable feedback you have provided. We have made our best effort to answer your questions and address your comments in the updated version of the manuscript. We kindly ask the reviewer to consider increasing their score. We are also happy to engage with the reviewer if they have any further questions.

---

### Author Response · Authors · 2025-11-22
**Response to all reviewers and ACs**

We would like to thank all the reviewers for their insightful and valuable feedback, which has helped indicate our contribution and improve our paper.

The reviewers have highlighted several strengths of our work:

- NW4M: “The MLIP training pipeline and architecture-agnostic contrastive loss function proposed in the paper are easy to follow… TransIP shows a significant improvement in force prediction compared to traditional data augmentation methods”
- emGi: “A friendly way to add equivalence. The core idea of this paper is clear. The latent equivariance loss is architecture-agnostic and straightforward to implement, allowing the backbone to remain unconstrained while satisfying equivariance by avoiding the use of spherical tensors.”
- RoKz: “The method shows comparable performance to state-of-the-art equivariant models and notably outperforms data augmentation baselines… Provides a scalable alternative to equivariant message-passing networks, highlighting potential for deployment on large molecular datasets.”
- RAy9: “The idea of learning latent equivariance through a contrastive objective is conceptually interesting and practically promising.”

Our work discusses an open question in molecular ML and MLIPs methods, where we can learn equivariance in unconstrained Transformer models.  In the revised manuscript, we have thoroughly addressed all comments and feedback from individual reviews. All revised texts in the main manuscript and supplementary materials have been highlighted in red to make them easy to find.

The updated version includes:

- **Additional results on the Open Molecules 2025 (OMol25) benchmark**: We would like to point out that OMol25 [1] is a diverse molecular dataset that covers 83 atomic elements and six primary categories: metal complexes, electrolytes, biomolecules, SPICE, neutral organics, and reactivity. We now explain this at the beginning of Section 5 in the revised version. We also include the dataset scaling results (1M, 2M, and 4M) comparing TransIP and TransAug for each category in Appendix B.2. Additionally,  we report the results for SPICE and reactivity in Table 7 in the revised version:


| Model    | SPICE Energy ↓ | SPICE Forces ↓ | Reactivity Energy ↓ | Reactivity Forces ↓ |
|-----------|:-------------:|---------------:|--------------------:|---------------------:|
| TransAug-S    | 11.5           | 151.3          | 23.0                | 179.7                |
| TransIP-S    | 8.7            | 121.8          | 17.8                | 136.4                |


We noticed these results are consistent with other molecule splits in the paper, where TransIP significantly outputs TransAug on both energy and force metrics.

- **Open Catalyst benchmark:** We have now included new evaluations on the Open Catalyst 2020 (OC20) benchmark [2].
We train on the 2M subset from the Structure-to-Energy-and-Forces (S2EF) task, and for validation, we selected $20,000$ samples from each validation split:  `val_id`  (in-distribution) and  `val_ood` (out-of-distribution).
We use the small version of TransIP and TransAug with the same hyperparameters in Tables 3 and 5 in the updated version of the paper, trained for 30,000 steps. We include the results in Table 6, which shows that TransIP outperforms TransAug on energy metrics on both in-distribution and out-of-distribution splits, and matches TransAug on force MAE.


| Model      | `val_id` Energy ↓ | `val_id` Forces ↓ | `val_ood` Energy ↓ | `val_ood` Forces ↓ |
|-----------|-------------------:|-------------------:|--------------------:|---------------------:|
| TransAug-S | 56                | 82                | 72                 | 95                  |
| TransIP-S  | 45                | 82                | 55                 | 95                  |

---

> ### Author Response · Authors · 2025-11-22
>
> - **Analysis on latent equivariance:**  We have included new results analyzing the effect of latent equivariance in embedding space. We ask whether the effect of rotating different inputs can be explained by a single group action in the latent space, i.e.,
> whether there exists a representation $\rho(g):\mathbb{R}^d\to\mathbb{R}^d$ such that
> $f\bigl(\phi(g)(m)\bigr)\approx\rho(g)f(m)$,
> where $f_\theta:\mathcal{M}\to\mathbb{R}^d$ denotes the embedding network, and $g\in\mathrm{SO}(3)$ acts on a molecule $m$ via the input representation $\phi(g)$ (rotation of atomic coordinates).
> Because $\rho(g)$ is unknown, we compute an approximate group action $\widehat{\rho}(g)\in\mathrm{O}(d)$ by solving an orthogonal Procrustes problem on
> embeddings from validation samples (obtained from a trained TransIP model).
> More details can be found in Section 7 in the revised manuscript.
> Our results show that the magnitude of the rotation-induced discrepancy of different molecules drops after applying the single orthogonal alignment, and we also found that the aligned channels of the embedding space match entrywise, concentrating along the identity. These results indicate that TransIP learns an embedding where input rotations act approximately as a shared orthogonal transformation, even though explicit equivariance was not enforced in the architecture.
>
>
> Our detailed point-by-point responses to each comment in the individual reviews are described below, respectively.
>
> References:
>
> 1. The Open Molecules 2025 (OMol25) Dataset, Evaluations, and Models. Levine et al., 2025.
> 2. The Open Catalyst 2020 (OC20) Dataset and Community Challenges. Chanussot et al., ACS 2021.

---

### Author Response · Authors · 2025-12-03
**Summary**

Dear Area Chair,

We are writing to provide an update on our submission and to express our appreciation for the reviewers' time and valuable feedback. We would like to thank all the reviewers for their time and comments on our paper, and ACs for their effort in this difficult situation.

Our paper addresses the problem of learning roto-translational symmetry in molecular ML applications. Our approach guides a generic non-equivariant Transformer-based architecture to learn symmetry by optimizing its representations in the embedding space. We evaluate our approach using the recent OMol25 benchmark and compare it with a Transformer trained with data augmentation as the main baseline.

In the updated manuscript, we have thoroughly addressed all comments and feedback from individual reviews provided during the rebuttal period. Unfortunately, the reviewers did not have the opportunity to meaningfully engage with our rebuttal before the cutoff. We provide below a brief summary of the changes in our paper resulting from the rebuttal:

**SPICE and Reactivity results from OMol25 benchmark**: As requested by reviewers NW4M and emGi, we included the results of the SPICE subset where our model TransIP significantly outperforms TransAug on both energy and force metrics.

**Open Catalyst benchmark**: As requested by reviewers NW4M and emGi, we have also included new evaluations on the Open Catalyst 2020 (OC20) benchmark, with 2M training samples from the Structure-to-Energy-and-Forces (S2EF) task. On this task, TransIP outperforms TransAug on energy metrics on both in-distribution and out-of-distribution splits, and matches TransAug on force MAE.

**Analysis on latent equivariance**: As requested by reviewer RoKz, we have included results analyzing the effect of latent equivariance in embedding space, which show that our model TransIP learns an embedding where input rotations act approximately as a shared orthogonal transformation. We would also like to note that reviewer RoKz raised their score to 6 during rebuttal before the cutoff.

**Inference speed of TransIP and equivariant baselines**: As requested by reviewer RAy9, we have included inference speed comparisons of our models against an eSEN baseline using the same hardware. Both TransIP’s small and medium versions are significantly faster than the eSEN baseline, while TransIP-L is slightly faster than eSEN.

Yours sincerely

The Authors

---

### Note · Authors · 2026-01-29

I have read and agree with the venue's withdrawal policy on behalf of myself and my co-authors.

---

### Meta-Review · Area_Chair_YYYo · 2026-01-04

**Summary:**

The paper proposes TransIP, a Transformer-based inter-atomic potential model that learns SO(3) equivariance via a contrastive loss in the latent space rather than relying on explicit equivariant architectures. The authors argue this approach improves scalability and efficiency compared to hard-wired equivariant networks. The model was evaluated on the OMol25 dataset, with additional benchmarks (SPICE, OC20) added during the rebuttal.

While the reviewers appreciated the architecture-agnostic nature of the method and the interesting concept of learning latent equivariance, the consensus leans towards rejection. The primary concern is that the model does not achieve parity with state-of-the-art (SOTA) equivariant baselines, with one reviewer noting a significant gap in force prediction accuracy. Although the authors conducted a comprehensive rebuttal by adding datasets and speed comparisons, the "comparable accuracy" claims were disputed, and the trade-off between efficiency and accuracy was not deemed sufficient by the majority of reviewers.

**Reviewer Concerns:**

Concerns Addressed by Rebuttal:

1. Lack of Latent Space Analysis: Reviewer RoKz requested an investigation into the latent space. The authors successfully addressed this by providing a Procrustes analysis showing that input rotations act as a shared orthogonal transformation in the embedding space. This convinced RoKz to raise their score.

2. Limited Datasets: Reviewers NW4M and emGi criticized the exclusive reliance on the OMol25 dataset. The authors addressed this by adding results for SPICE and Open Catalyst 2020 (OC20).

3. Inference Speed: Reviewer RAy9 requested concrete inference speed comparisons. The authors provided these, showing TransIP is faster than the eSCN baseline on identical hardware.



Outstanding Concerns:

1. Performance Gap vs. SOTA: This remains the critical outstanding issue. Reviewer RAy9 noted that while TransIP beats data-augmentation baselines, it trails significantly behind SOTA equivariant models like eSCN in force Mean Absolute Error (MAE) (~0.10 eV/Å vs ~0.01 eV/Å). Reviewer NW4M also noted that TransIP generally does not outperform eSCN or GemNet-OC. The authors' counter-argument regarding training epochs and speed did not satisfy Reviewer RAy9, who maintained that the accuracy difference is too large to be termed "comparable".






2. Missing Strong Baselines: Reviewer RAy9 pointed out the omission of recent frontier models such as MACE, ViSNet, and SO3Krates. The authors argued that eSCN and GemNet-OC are sufficient SOTA representatives, but this did not resolve the reviewer's concern regarding the positioning of the paper against the current state of the field.



3. Generalizability of the Module: Reviewer emGi suggested applying the implicit-equivariance module to other invariant backbones to prove its general utility. The authors acknowledged this but left it for future work, leaving the method's broader applicability on non-Transformer architectures unverified.

**Reviewer Scores:**

Reviewer RoKz: 6 (Accept). This reviewer actively engaged in the discussion and raised their score from 4 to 6 after the authors provided the requested latent space analysis.


Reviewer RAy9: 4 (Reject). This reviewer explicitly stated that the response did not address their concerns regarding the accuracy gap and baseline comparisons and maintained their original rating.

Reviewer NW4M: 4 (Reject). While the authors provided the requested additional datasets (SPICE/OC20), the reviewer did not update their score. Given that their primary weakness was "TransIP does not perform well" compared to eSCN , and the rebuttal confirmed TransIP is "closer" but still generally inferior or requiring trade-offs, it is unlikely this reviewer would have moved above a borderline score.


Reviewer emGi: 4 (Reject). Similar to NW4M, this reviewer requested new datasets and ablation on other backbones. The datasets were provided, but the backbone ablation was deferred to future work. It is probable this reviewer would have maintained their score or moved only to a weak borderline, as the core contribution's generalizability wasn't fully demonstrated.

---

### Decision · Program_Chairs · 2026-01-26

Reject